# Culicoides jiangchengensis, a new species of the subgenus Sinocoides (Diptera, Ceratopogonidae) based on integrative taxonomy from China

Yuwen He[1], Jinxin Meng[1], Nan Li[1], Zhao Li[2], Taoying Yu[3], Laxi Zhang[2], Dongmei Wang[2], Guoping Liu[4]*, Jinglin Wang[1]*

1 Yunnan Tropical and Subtropical Animal Viral Disease Laboratory, Key Laboratory of Transboundary Animal Diseases Prevention and Control (Co-construction by Ministry and Province), Ministry of Agriculture and Rural Affairs, Yunnan Animal Science and Veterinary Institute, Kunming, China, 2 Jiangcheng County Animal Disease Prevention and Control Center, Jiangcheng, China, 3 Gongshan County Animal Disease Prevention and Control Center, Gongshan, China, 4 Center for Disease Control and Prevention of Shenyang Command, Shenyang, China

* Wangjl107@163.com (JW); kqhxj2@163.com (GL)

**Data Availability Statement:** All sequence are available from the accession number(s) OL471017-OL471027.

## Abstract

Biting midges of the genus *Culicoides* are important in both medicine and veterinary medicine because their blood-feeding regime enable them to transmit a variety of pathogens. In this study, the morphological characteristics of the new species of *Culicoides* (*Sinocoides*) *jiangchengensis* Wang et Liu sp. nov are described and compared with the other species of female *Culicoides* in the subgenus *Sinocoides*. Three morphological characteristics of *C. jiangchengensis*, such as without sensory pit in 3$^{rd}$ palpus segment, sensilla coeloconica on flagellomeres 1,9–13, and m$_1$ and m$_2$ cell of the wings with pale spots, were different from the other nine species of culicoides in subgenus *Sinocoides*. Genetically, *C. jiangchengensis* are most closely related to *C. malipoensis*, but they were located in different branches and the minimum interspecific distance between them was 12.6%. In addition, a checklist of 10 species in the subgenus *Sinocoides* Chu, 1983 (Diptera: Ceratopogonidae: *Culicoides*) in China, including the new species *C. jiangchengensis* Wang et Liu sp. nov., is provided, and an updated key to species of the subgenus *Sinocoides* Chu, 1983 was presented.

## Introduction

Biting midges (Ceratopogonidae) are important vector insects that transmit arboviruses causing serious disease in human or animal, such as Oropouche virus, bluetongue virus (BTV), epizootic haemorrhagic disease virus (EHDV), African horse sickness virus (AHSV), Ibaraki disease virus (IBADV), Schmallenberg virus (SBV), Bovine ephemeral fever virus (BEFV), and vesicular stomatitis virus (VSV) by biting humans, livestock, poultry, and other animals [1–3]. Currently, 1764 species of bloodsucking midge are classified in four genera worldwide, of which *Culicoides* Latreile (Diptera, Ceratopogonidae) is the largest, including 1399 (79.31%)

**Funding:** This work is supported by grants from the Basic Research Projects of Yunnan Province (grant no. 2019FA015 and 202201AS070062), National Natural Science Foundation of China (grant no. 32260896), Scientific and technological innovation team construction project of Kunming Medical University (grant no. CXTD202111), Projects funded by the central government to guide local scientific and Technological Development (202207AB110006), and Yunnan Chenggong expert workstation (202005AF150034). The funders had no role in the study design, data collection and analysis, decision to publish, or preparation of the manuscript.

**Competing interests:** The authors have declared that no competing interests exist.

species in 38 species groups in 33 subgenera [4]. In China, there are 451 species of bloodsucking midges in three genera, and *Culicoides* includes 336 species in 12 subgenera [5,6]. Qu (1983) established the subgenus *Sinocoides* Chu, 1983 in genus *Culicoides* [7]. At that time, only one species of this subgenus was described in China, *Culicoides hamiensis* [8]. Subsequently, eight new species in the subgenus were described by Liu, Yu et al [9–11]. In the last decade, Liu and Wu et al. described four new species in this subgenus [12–14]. Here, a new species of the subgenus *Sinocoides* is described in an investigation on blood-sucking midges in Yunnan Province, China. Cytochrome c oxidase subunit gene (COI) sequences were obtained to identify the species of *Culicoides*. The main distinguishing characteristics, geographical distribution, and a species checklist of the 10 species of this subgenus found in China are reported.

## Materials and methods

Biting midges were collected overnight from 7:00 pm to 7:00 am the next morning in domestic animal pens using light traps (12 V, 300 mA; Wuhan Lucky Star Environmental Protection, Hubei, China) in Jiangcheng County in September 2015 and Gongshan County in August 2017 of Yunnan Province, China. The midges were stored in 70% ethanol at 4°C and immersed in 250 μL tissue digestive solution with 1% proteinase K (TIANGEN DNA extraction kit) for non-destructive tissue digestion [15]. The midges were slide-mounted in Canada balsam, as described by Yu et al [5].

Midge genomic DNA was extracted from the digestive supernatant using Micro DNA Kit (TIANGEN, Beijing, China) according to the manufacturer's instructions. Partial mitochondrial cytochrome c oxidase I (COI) gene sequence was obtained by PCR amplification using forward primer C1-J-1718 (5'-GGAGGATTTGGAAATTGATTAGT-3') and reverse primer C1-N-2191 (5'-CAGGTAAAATTAAAATATAAACTTCTGG-3') [16]. The PCR reaction volume was 50 μL, and contained *Takara Ex Taq* (5 U/μL) 0.25 μL, 10 × *Ex Taq* Buffer 5 μL, dNTP Mixture (2.5 mM) 4 μL, 0.5 μM of each primer and 4 μL of midge genomic DNA. The DNA amplification program was: 94°C 3 min, 30 cycles of 94°C 30 s, 55°C 30 s, 72°C 30 s, followed by 72°C 5 min. Purified amplicon of the COI gene was cloned into the pMD™19-T vector (Takara, Dalian, China). Recombinant plasmids were transformed into *Escherichia coli DH5α* competent cells. Positive clones were identified through PCR using M13 universal primers and sequenced using an automated ABI 3730 DNA Sequencer (Applied Biosystems). The COI gene sequence was submitted to GenBank under accession number OL471017-OL471027. Sequence alignments were performed using Clustal X (version 2.0) [17] and MAFFT [18] to ensure proper alignment. Phylogenetic trees were constructed by the neighbour-joining method using distance matrices generated by the p-distance determination algorithm in MEGA-X with 1000 bootstrap replicates.

### Nomenclatural acts

The electronic edition of this article conforms to the requirements of the amended International Code of Zoological Nomenclature, and hence the new names contained herein are available under that Code from the electronic edition of this article. This published work and the nomenclatural acts it contains have been registered in ZooBank, the online registration system for the ICZN. The ZooBank LSIDs (Life Science Identifiers) can be resolved and the associated information viewed through any standard web browser by appending the LSID to the prefix "http://zoobank.org/". The LSID for this publication is: urn:lsid:zoobank.org:pub:E8649BF5-471C-4812-A61E-360486C59946. The electronic edition of this work was published in a

journal with an ISSN, and has been archived and is available from the following digital repositories: PubMed Central, LOCKSS.

## Ethics statement

Authorization for the collection of Culicoides has been obtained from Institute for Yunnan Animal Science and Veterinary Institute, Kunming, China (protocol approval number: 2019FA015 and 202005AF150034). No specifc permits were required for the field studies. After explanation of the purposes and activities of the study, oral consent was obtained from the local participating residents prior to Culicoides collection. No sites were protected by law and this study did not involve endangered or protected species.

## Results

### Taxonomy

*Culicoides* (*Sinocoides*) *jiangchengensis* **Wang et Liu sp. Nov, 2023. (Fig 1).** urn:lsid:zoobank.org:act:C059685C-7A56-4CD2-9A67-946CF68C2A67

**Diagnosis.** Females: only for *Culicoides* species with the following combination of features: palpus third segment slightly swollen at distal 1/3, without sensory pit, with sensilla

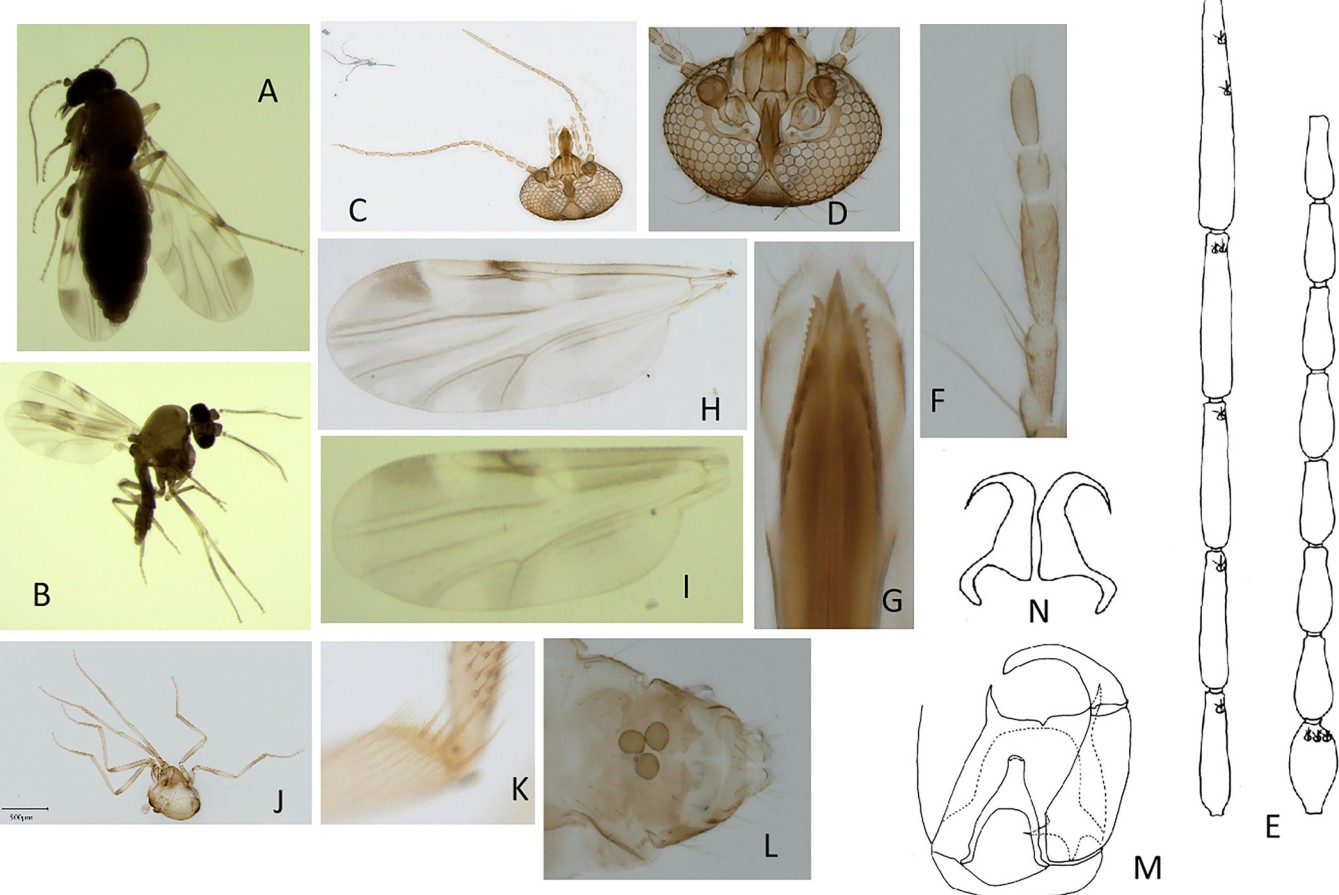

**Fig 1. *Culicoides* (*Sinocoides*) *jiangchengensis* Wang et Liu, sp. nov.** A: Female; B: Male; C: Head; D: Front; E: Antenna; F: Maxillary palpus; G: Mandibles; H: Wing (Female); I: Wing (Male); J: Thorax; K: Hind tibial comb; L: Spermatheca; M: Hypopygium; N: Parameres.

scattered on the surface; a small, approximately oval, fuzzy pale spot at the distal end of $r_5$ cell; $m_1$ with two pale spots, anal cell with a wide pale band extended from its base to distal; three equally sized spermathecae. Male: only for *Culicoides* species in Yunnan with the following combination of features: palpus third segment without sensory pit; anal cell with a big, long irregular pale spot shaped like a pocket; the arch of aedeagus with deep bow type, the middle of aedeagus reflexed in apex, parameres separate, the apex part bends in a hook shape.

## Description

**Female.   Body** (Fig 1A) Small and medium-sized midges. Wing length 1.08 (1.0–1.22) mm, breadth 0.48 mm, CR 0.50 (0.48–0.53, n = 9).

**Head** (Fig 1C, 1D, 1E, 1F and 1G) Brown. Eyes bare, separated by distance approximately equal to the diameter of one ommatidium, and a transverse seam in the lower forehead. Antenna pedicel slightly pale brown; flagellomeres 1–8 short and stout, flagellomeres 9–13 longer, lengths of flagellomeres in the proportion of 19:19:19:19:19:19:18:19:26:28:34:35:55; AR 1.13 (1.08–1.18, n = 9); sensilla coeloconica on flagellomere 1, 9–13. Palpus five-segmented, lengths in the proportion of 10:20:22:12:13, segment 3 slightly swollen at apical 1/3, without sensory pit, with sensilla scattered on the surface; PR 2.63 (2.50–2.75, n = 9). Three bristles on each side of the frontoclypeus. Proboscis length 150 μm, head height 250 μm, P/H ratio 0.59 (0.53–0.64, n = 9); mandible with 15–17 teeth (n = 9), maxilla with 14 teeth (n = 9).

**Thorax** (Fig 1J and 1K) Scutum light brown, scutellum dark brown; wing with contrasting pattern of pale/dark spots; wing base with a large pale spot and connected to the wide pale band of the anal cell; basal 2/3 of 1st radial cell and r-m cross-vein in pale spot, and extension posterior over M1 to connect with narrow pale band of $m_2$; distal 4/5 of 2nd radial cell covered by pale spot; apical pale spot in cell $r_5$ cell faint, and reaching wing margin; $m_1$ with two pale spots, proximal pale spot long band; $m_2$ with two differently shaped pale spots: proximal pale spot long band extending from the base to distal, distal pale spot abutting the wing margin; $m_4$ with a pale spot abutting wing margin; anal cell with a wide pale band extending from base to distal; macrotrichia on the costal of the proximal of $r_5$ cell and distal of $m_1$, but no on the proximal of $m_1$; hind tibial comb with four spines, 2nd spine longest, metatibial comb (mc) about 18 teeth, TR and F-T of legs are as in Table 1.

**Abdomen** Light brown. Three subequal-size ovoid spermathecae (Fig 1L), each measuring 25.00×23.00 μm.

**Male** (Fig 1B) Similar to female with usual sexual differences.

**Head** The lengths of Antenna flagellomeres in the proportion 26:13:13:14:14:14:12:11:10:9:34:33:38, sensilla coeloconica on flagellomeres 1, 11–13; palpus with segments in proportions of 8:10:11:7:9, PR 1.57 (n = 1); one bristles on each side of the frontoclypeus.

**Thorax** Wing with pattern of pale spots as in Fig 1B, wing length 0.82 (n = 1), width 0.33 (n = 1); CR 0.63 (n = 1); wings light dark, with eight pale spot; wing costal with three obvious

**Table 1.  Tarsal ratios (TR) and measurements of leg segments and tarsomeres from femur to tarsomere 5 (F-T) of all legs of *Culicoides* (*Sinocoides*) *jiangchengensis* Wang et Liu sp. nov.**

|  | Leg | TR | F-T |
|---|---|---|---|
| Female(♀) | Foreleg | 2.71 | 78:78:38:14:10:8:10 |
|  | Midleg | 2.89 | 97:101:52:18:12:13:9 |
|  | Hindleg | 1.68 | 92:95:49:23:15:9:10 |
| Male(♂) | Foreleg | 2.42 | 58:51:29:12:9:8:9 |
|  | Midleg | 3.17 | 72:67:38:12:10:7:8 |
|  | Hindleg | 2.07 | 66:65:31:15:10:8:9 |

dark spots; basal 1/3 of 1st radial cell and r-m cross-vein in pale spot; distal 2/3 of 2nd radial cell covered by pale spot; anal cell with a big irregular pale spot. Metatibial comb (mc) about 18 teeth, TR and F-T of legs are given as Table 1.

**Genitalia** (Fig 1M and 1N): the middle of the posterior margin of the 9th genitalia sternite concave, wide, arc and the membrane free of microhairs. The posterior margin 9th basal tergite flat, a small V-shaped concave was observed in the middle part, parameres robust, sharp Angle shape. Gonocoxite slender in its basal part, the dorsal ankle digitation, mucro part bend to the medial side. The middle aedeagus is nearly tapered, aedeagus with deep bow type, the arch height of the aedeagus is about 1/2 of the total length of the aedeagus, reflexed in apex. Parameres separated, the middle part thick, diminution, curve and with a hook shape in apex.

**Etymology.** The name jiangchengensis refers to the collecting location of the species.

**Type material.** Holotype female, Qiaotouhe Village, Menglie Town, Jiangcheng County, Yunnan Province, China (22°54′49″N, 101°88′70″E), September 2015, light trap. Male, Maxidang Village, Bangdang Town, Gongshan County, Yunnan Province, China (27°49′54.8328″N, 98°41′46.0536″E), Augest 2017, light trap.

**Paratypes:** 9 females and 1 male, same data as holotype.

**Distribution.** China (Jiangcheng County and Gongshan county of Yunnan Province).

**Remarks.** *C. jiangchengensis* collected in Jiangcheng County in this study have three spermathecae, which were similar to those of the subgenus *Pontoculicoides*, *Sinocoides*, *Jilinocoides* and *Trithecides*. Two eyes of *C. jiangchengensis* were separated, excluding the subgenus *Jilinocoides* and *Trithecides*, and the wings have pale spots and dark spots excluding *pontoculicoides*, indicating that *C. jiangchengensis* is a member of the subgenus *Sinocoides*. At present, there are 9 species of *Culicoides* in subgenus *Sinocoides*, among which *C. anthropophygas* (Fig 2), *C. hamiensis* (Fig 3), *C. jinghongensis* (Fig 4), *C. kongmiaoensis* (Fig 5), *C. multifarious* (Fig 6), *C. opertus* (Fig 7) had an obvious sensory pit in the 3rd segment of palpus, while *C. jiangchengensis* had no an obvious sensory pit, but with capitate sensilla scattered on the surface, which had significant differences between them. Although *C. pungobovis* (Fig 8) and *C. jiangchengensis* have similar morphological characteristics in the 3rd segment of palpus, their sensilla coeloconica on flagellomere are significantly different, the former is 1,6–8, while the latter is 1,9–13. *C. malipoensis* (Fig 9), *C. nanniwanensis* (Fig 10) and *C. jiangchengensis* had similar morphological characteristics in the 3rd palpus segment and the sensilla coeloconica on flagellomere, but there was no pale spot in the wings $m_1$ and $m_4$ of *C. nanniwanensis*, while the $m_1$ and $m_4$ of *C. malipoensis* and *C. jiangchengensis* with pale spot. *C. jiangchengensis* is the most similar to *C. malipoensis* in morphological characteristics, but the former has 2 pale spots in the $m_1$ and a wide pale band extended from the basal to the distal in the anal cell, while the latter has a pale band extended from base to the distal in the $m_1$ and a pale spot at the proximal in the anal cell.

In addition, the main characteristics distinguishing *Culicoides* in China are wing length, costal ratio (CR), antennal ratio (AR), proboscis ratio (PR), mandible teeth, and sensilla coeloconica on flagellomere. Table 2 provides detailed measurements.

## DNA analyses

The phylogenetic tree based on the COI gene sequences of *C. jiangchengensis*, *C. malipoensis* from Jiangcheng County and Gongshan County, Yunnan Province and another 24 species of *Culicoides* shows that nine female *C. jiangchengensis* from Jiangcheng and two male *C. jiangchengensis* from Gongshan formed a separate branch (Fig 11). The mean intraspecific distance was 1%, and the maximum was 2.36%. Although *C. jiangchengensis* and *C. malipoensis* are on different branches, their genetic relationship is closest among other *Culicoides*; the minimum interspecific distance was 12.6%.

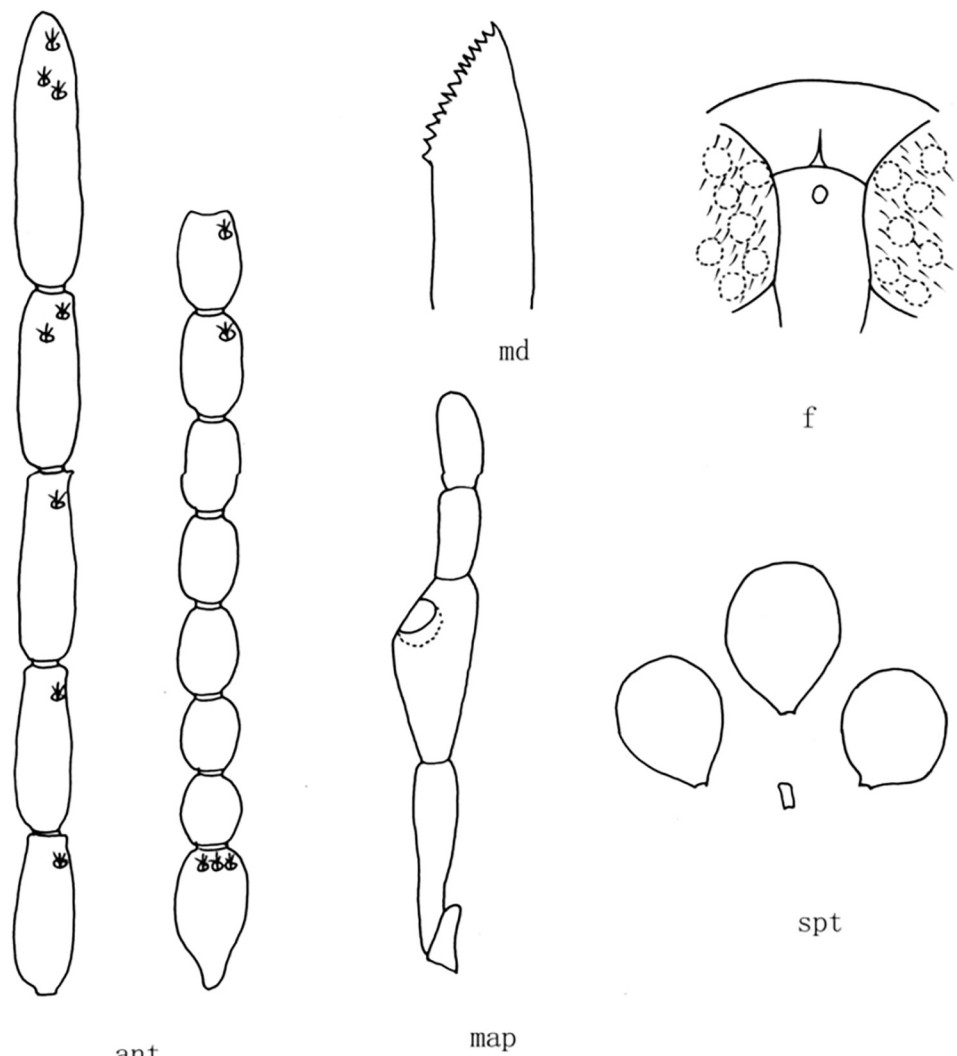

**Fig 2. *Culicoides* (*Sinocoides*) *anthropophygas* Yu et Liu, 2005 [5].** ant, antenna; f, front; map, maxillary palpus; md, mandibles; spt, spermatheca; w, wing.

## Checklist of *Sinocoides* in China and their geographical distribution

There are 10 species in the subgenus *Sinocoides* in China, distributed in eight provinces and autonomous regions of China.

(1) *Culicoides* (*Sinocoides*) *jiangchengensis* Wang et Liu sp. nov. (Fig 1) Type locality: China: Yunnan (Jiangcheng).

   Distribution: Yunnan (Jiangcheng, Gongshan).

(2) *Culicoides* (*Sinocoides*) *anthropophygas* Yu et Liu, 2005 [5] (Fig 2)

   *Culicoides* (*Sinocoides*) *anthropophygas* Yu et Liu, 2005 [5]:892; Type locality: China: Sichuan (Jiulong).
   Distribution: Sichuan (Jiulong).

(3) *Culicoides* (*Sinocoides*) *hamiensis* Chu, Qian et Ma, 1982 [8] (Fig 3)

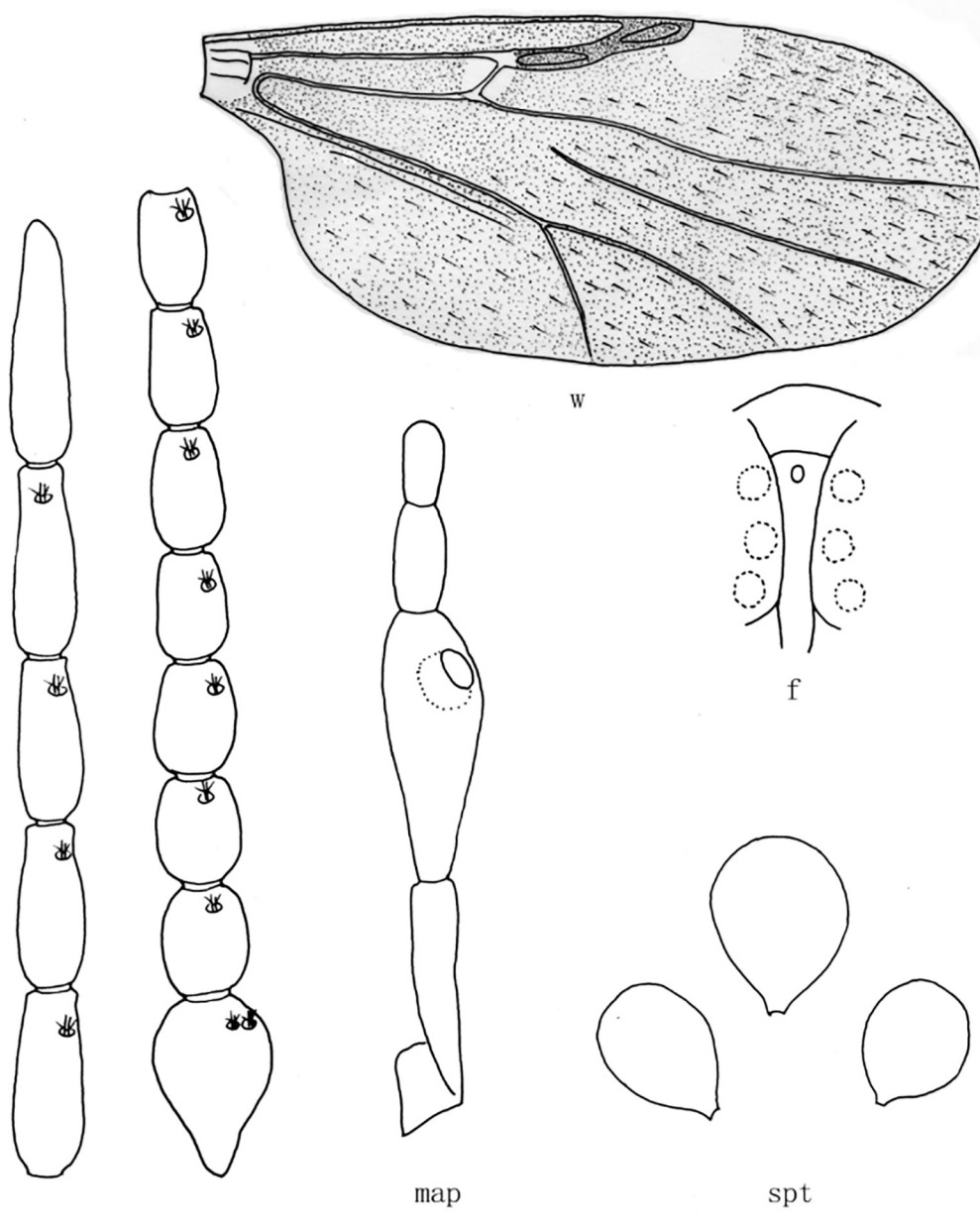

**Fig 3. *Culicoides* (*Sinocoides*) *hamiensis* Chu, Qian et Ma, 1982 [8].** ant, antenna; f, front; map, maxillary palpus; md, mandibles; spt, spermatheca; w, wing.

*Culicoides* (*Sinocoides*) *hamiensis* Chu, Qian et Ma, 1982 [8]:105; Type locality: China: Xinjiang (Hami).

Distribution: Xinjiang (Hami).

(4) *Culicoides* (*Sinocoides*) *jinghongensis* Wu et Liu, 2018 [14] (Fig 4)

*Culicoides* (*Sinocoides*) *jinghongensis* Wu et Liu, 2018 [14]:290; Type locality: China: Yunnan (Jinghong).

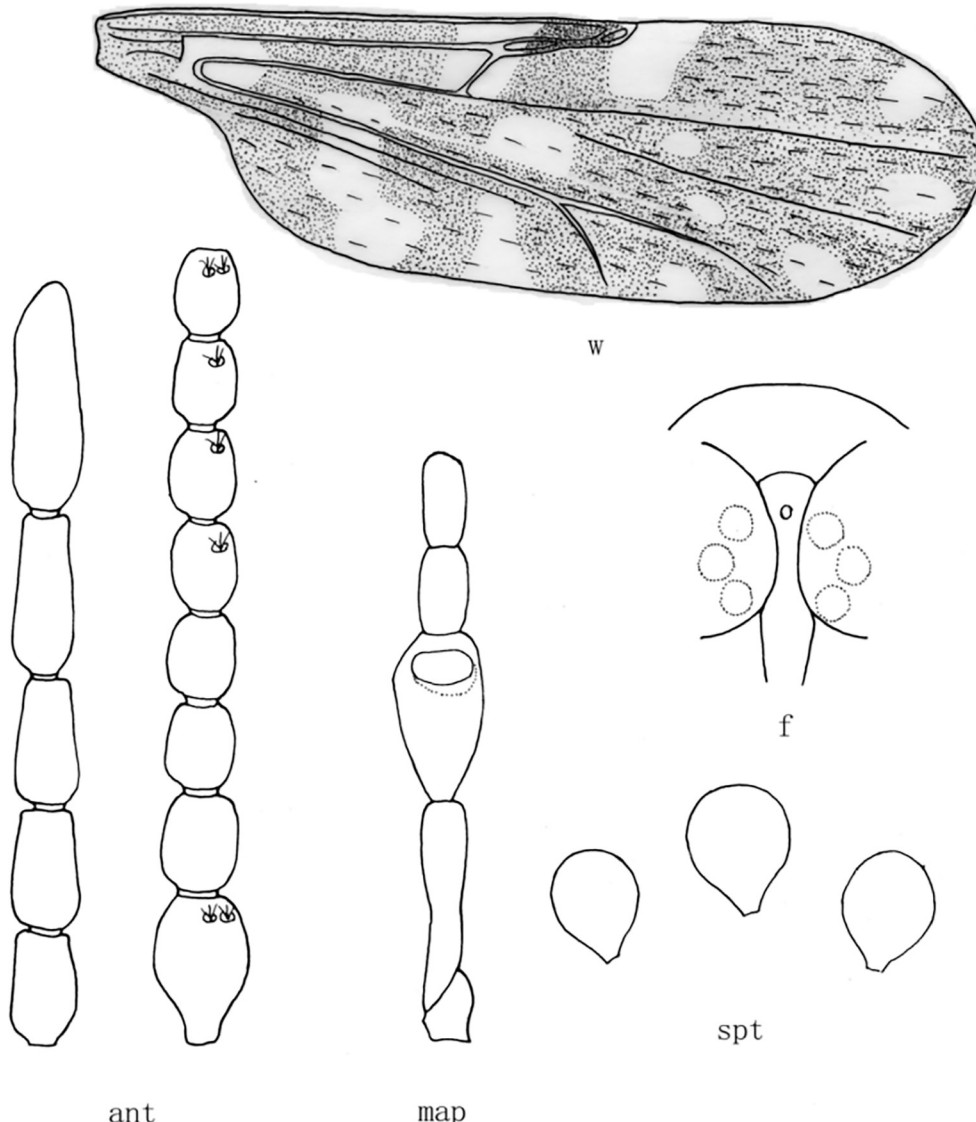

**Fig 4. *Culicoides* (*Sinocoides*) *jinghongensis* Wu et Liu, 2018 [14].** ant, antenna; f, front; map, maxillary palpus; md, mandibles; spt, spermatheca; w, wing.

Distribution: Yunnan (Jinghong).

(5) *Culicoides* (*Sinocoides*) *kongmiaoensis* Liu et Zhou, 2006 [12] (Fig 5)

*Culicoides* (*Sinocoides*) *kongmiaoensis* Liu et Zhou, 2006 [12]: 467; Type locality: China: Shandong (Qiuhu).
Distribution: Shandong (Qiuhu).

(6) *Culicoides* (*Sinocoides*) *multifarious* Liu, Gong et Zhang, 2003 [11] (Fig 6)

*Culicoides* (*Sinocoides*) *multifarious* Liu, Gong et Zhang, 2003 [11]:359; Type locality: China: Gansu (Tianshui).
Distribution: Gansu (Tianshui).

(7) *Culicoides* (*Sinocoides*) *opertus* Liu et Yu, 1990 [9] (Fig 7)

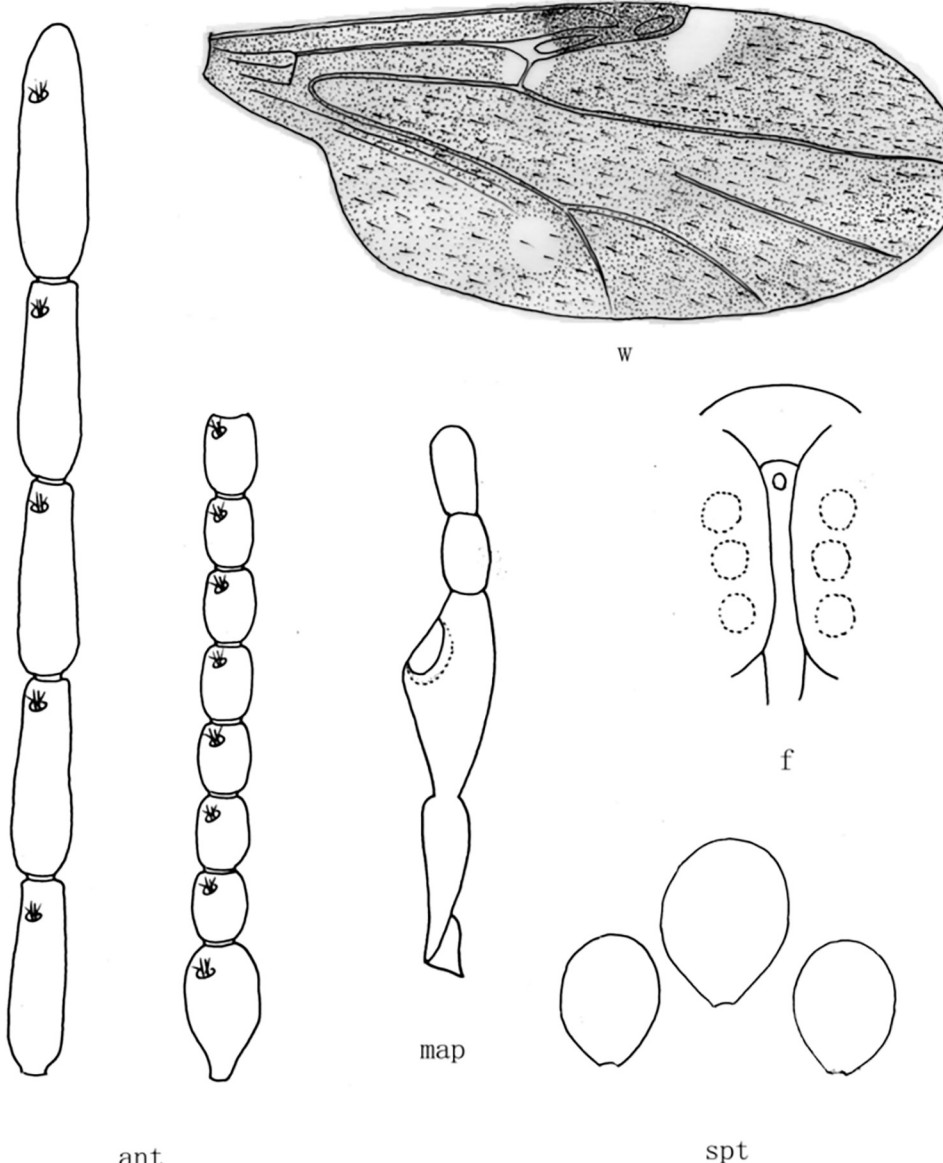

**Fig 5. *Culicoides* (*Sinocoides*) *kongmiaoensis* Liu et Zhou, 2006 [12].** Ant, antenna; f, front; map, maxillary palpus; md, mandibles; spt, spermatheca; w, wing.

*Culicoides* (*Sinocoides*) *opertus* Liu et Yu, 1990 [9]:15; Type locality: China: Heilongjiang (Hulin).

Distribution: Heilongjiang (Hulin, Raohe, Suifenhe).

(8) *Culicoides* (*Sinocoides*) *pungobovis* Liu, Yan et Liu, 1996 [10] (Fig 8)

*Culicoides* (*Sinocoides*) *pungobovis* Liu, Yan et Liu, 1996 [10]:35; Type locality: China: Hainan (Qiongzhong).

Distribution: Hainan (Qiongzhong).

(9) *Culicoides* (*Sinocoides*) *malipoensis* Liu et Ren, 2011 [13] (Fig 9)

*Culicoides* (*Sinocoides*) *malipoensis* Liu et Ren, 2011 [13]:257; Type locality: China: Yunnan (Malipo).

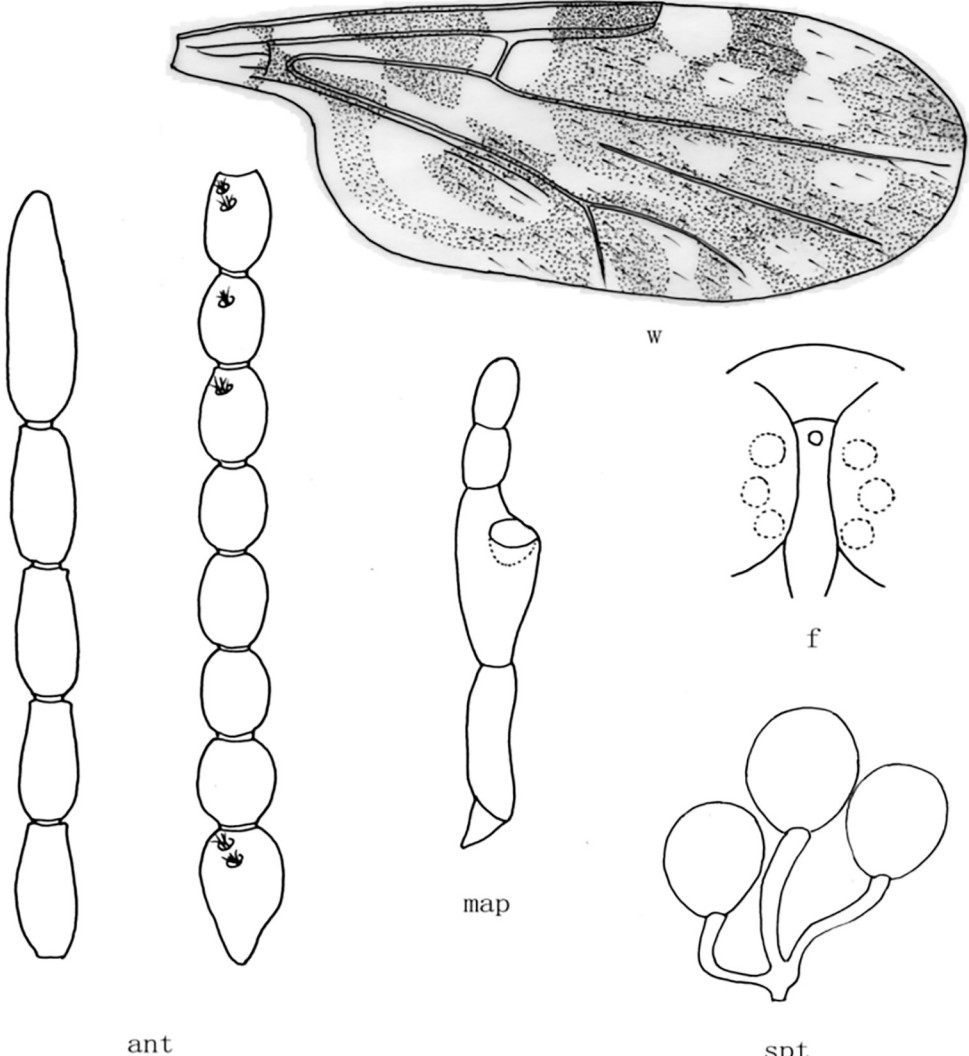

**Fig 6. *multifarious* Liu, Gong et Zhang, 2003 [11].** ant, antenna; f, front; map, maxillary palpus; md, mandibles; spt, spermatheca; w, wing.

Distribution: Yunnan (Malipo)

(10) *Culicoides* (*Sinocoides*) *nanniwanensis* Liu et Wang, 2011 [13] (Fig 10)

*Culicoides* (*Sinocoides*) *nanniwanensis* Liu et Wang, 2011 [13]:257; Type locality: China: Shanxi (Nanniwan).
Distribution: Shanxi (Nanniwan).

## Discussion

The three main morphological characteristics that identified *C. jiangchengensis* as subgenus *Sinocoides* are the separation of two eyes, capitate sensilla scattered on the surface of 3rd segment palpus, the contrasting pattern of pale/dark spots in the wing, and three subequal-size ovoid spermathecae, which is similar to that of other species of subgenus *Sinocoides*.

According to the following three morphological characteristics of *C. jiangchengensis*: 1) without sensory pit in 3rd segment of palpus; 2) sensilla coeloconica on flagellomere 1, 9–13; 3)

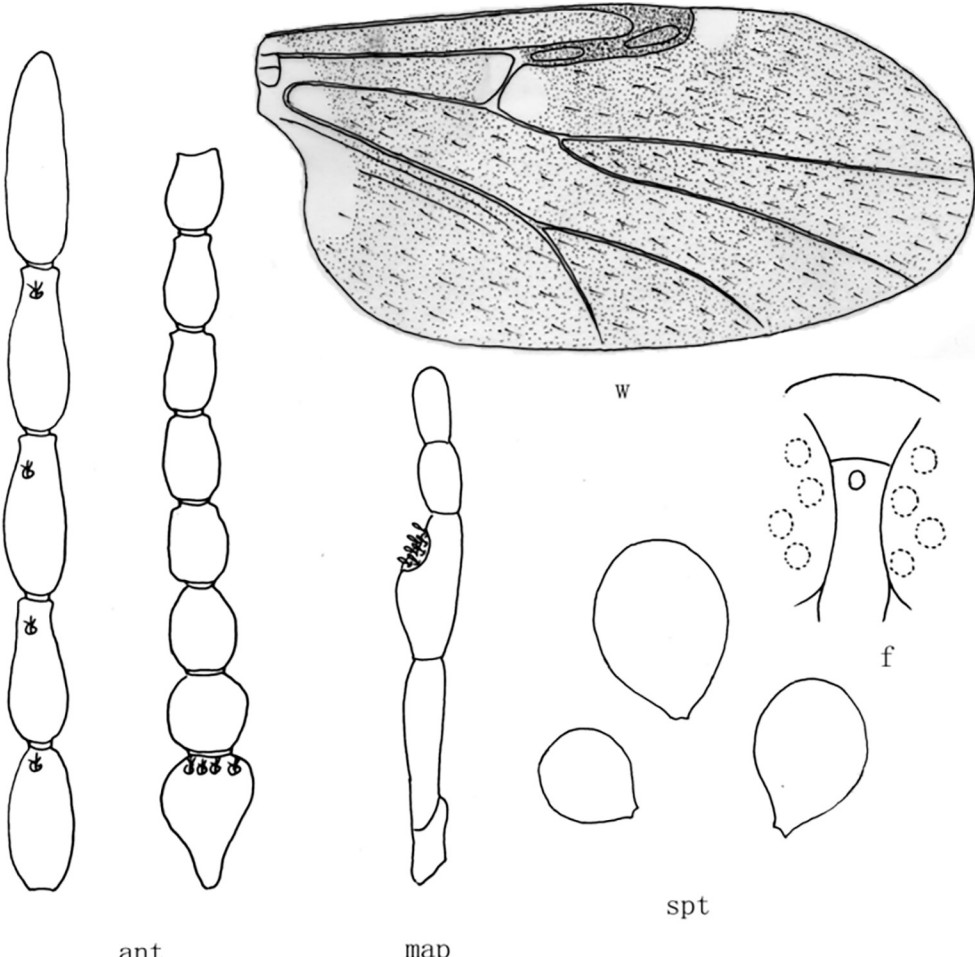

**Fig 7. *Culicoides* (*Sinocoides*) *opertus* Liu et Yu, 1990 [9].** ant, antenna; f, front; map, maxillary palpus; md, mandibles; spt, spermatheca; w, wing.

$m_1$ and $m_2$ cell of the wings with pale spots, which were different from the other eight species of *culicoides* in subgenus *Sinocoides*, such as *C. anthropophygas*, *C. hamiensis*, *C. jinghongensis*, *C. kongmiaoensis*, *C. miaoensis*, *C. opertus*, *C. pungobovis* and *C. nanniwanensis*. *C. jiangchengensis* is morphologically similar to *C. malipoensis*, but *C. malipoensis* is a medium-sized *Culicoides*, with a wing length of 1.35 mm, mandible with nine teeth, P/H ratio 0.64, the distal of $r_3$ with a large fuzzy pale spot, $m_1$ cell with a wide pale band extending from the base to distal, and distal of anal cell with a pale spot, which is obviously different from *C. jiangchengensis*.

Molecular biology is one of the methods for rapid and accurate identification of vector insect species [19]. COI gene with moderate evolutionary rate is the most commonly used molecular target for mosquito and midges identification [19,20]. In this study, the COI gene was used as a molecular target to identify *C. jiangchengensis* and *C. malipoensis* collected in Jiangcheng and Gongshan. The results showed that *C. jiangchengensis* was mostly related to, but distinct from, *C.malipoensis* in subgenus *Sinocoides*. The minimum interspecific distance between them was 12.6%, higher than between *C. selandicus* and *C. kalix* (5.9%) [21] and *C. fagineus* F1 and *C. subfagineus* (*sensu stricto*) (12%) within the subgenus *Culicoides* [22] and *C. bolitinos* and *C. tutti-frutti* (9.5%) within the subgenus *Avaritia* [23] (Augot et al. 2016). These data indicated that *C. jiangchengensis* is a new species in subgenus *Sinocoides* based on its

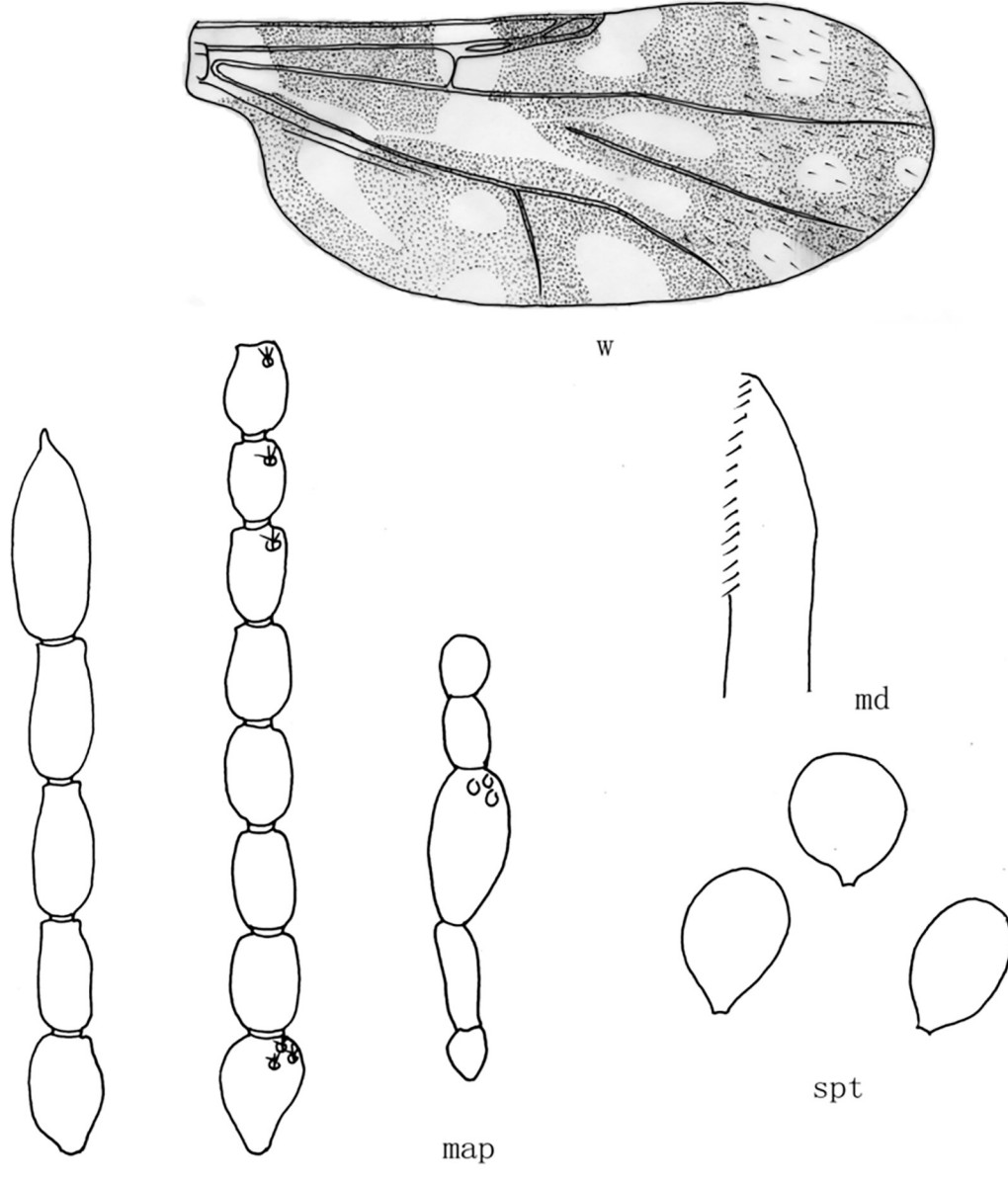

**Fig 8. *Culicoides* (*Sinocoides*) *pungobovis* Liu, Yan et Liu, 1996 [10].** ant, antenna; f, front; map, maxillary palpus; md, mandibles; spt, spermatheca; w, wing.

morphology and molecular biology, increasing the number of *Culicoides* species in China. Meanwhile, this is the first time to obtain the sequence of *Culicoides* in subgenus *Sinocoides*, which provides sequence information for rapid molecular identification and genetic evolution of *Culicoides* species in China.

Since subgenus *Sinocoides* was established in China in 1983, females of nine *Culicoides* species in this subgenus have been recorded in different regions of China. This survey discovered a new *Culicoides* species in *Sinocoides* in Yunnan, increasing the species in this subgenus to 10. Based on the literature, a key to the females of 10 *Culicoides* in subgenus *Sinocoides* was

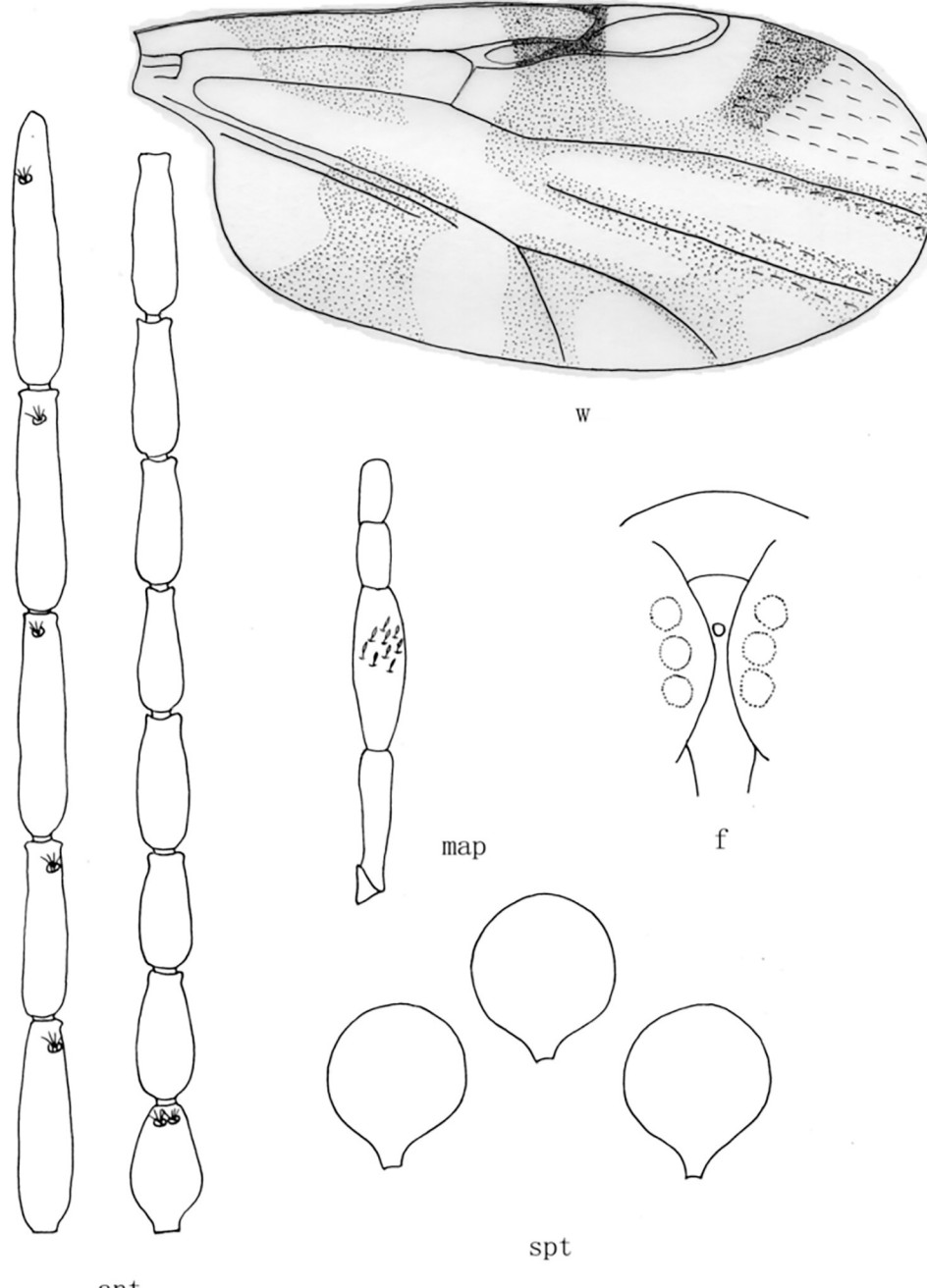

**Fig 9. *Culicoides* (*Sinocoides*) *malipoensis* Liu et Ren, 2011 [13].** ant, antenna; f, front; map, maxillary palpus; md, mandibles; spt, spermatheca; w, wing.

compiled. Fortunately, in this study, we used molecular biology methods to identify the males of *C. jiangchengensis*, and systematically described the morphological characteristics of the males, providing data for the classification and identification of *C. jiangchengensis* in the future. However, due to the lack of morphological description of other subgenus *Sinocoides* male, failed to compile a key to the males of 10 *Culicoides* in this subgenus, and needs further investigation.

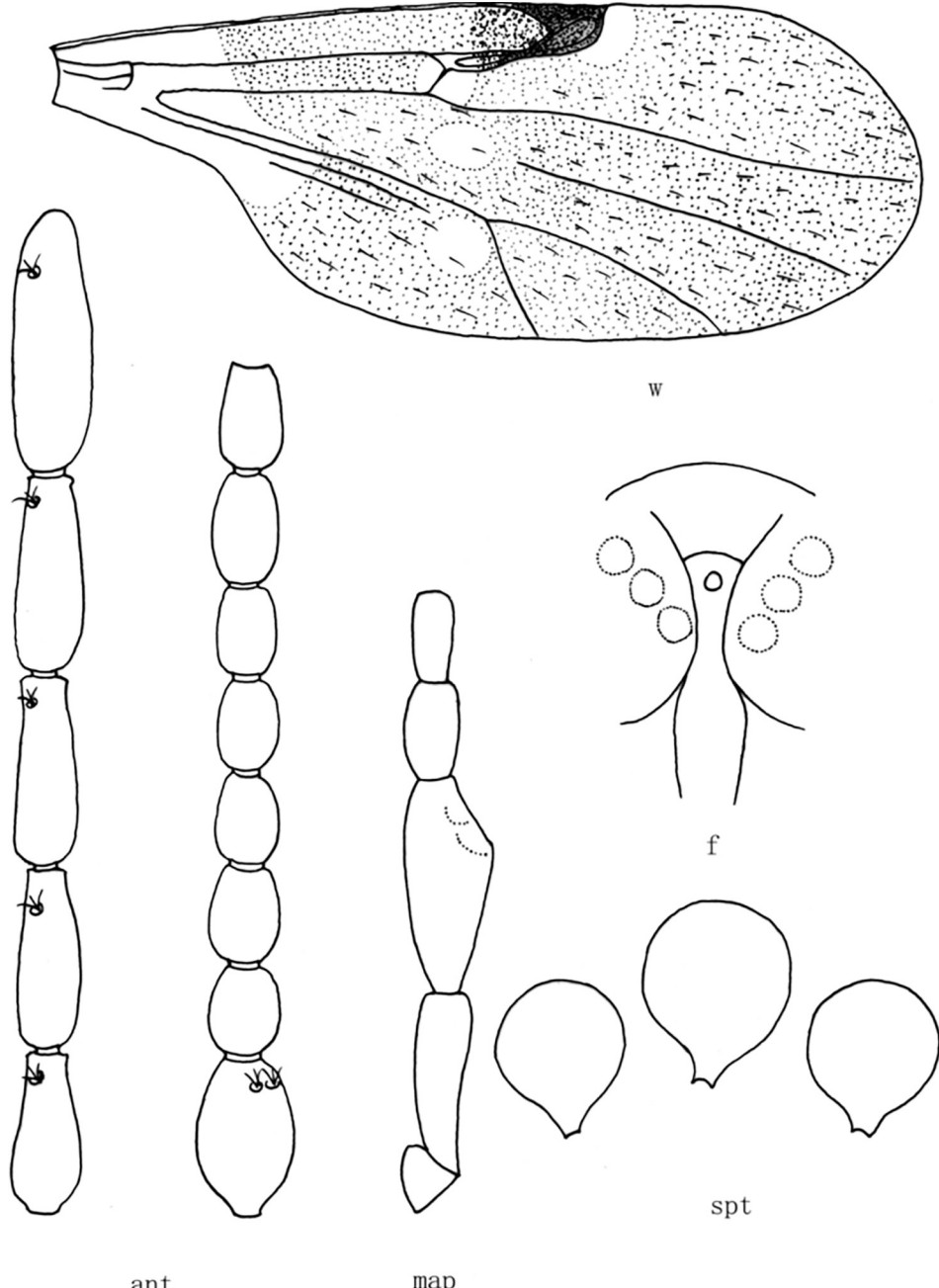

**Fig 10.** *Culicoides* (*Sinocoides*) *nanniwanensis* **Liu et Wang, 2011 [13].** ant, antenna; f, front; map, maxillary palpus; md, mandibles; spt, spermatheca; w, wing.

## An updated key to females in the *subgenus Sinocoides* of *Culicoides*

1 Eyes hairy, Wing without pale and dark spot (Fig 2)...... *C.* (*S.*) *anthropophygas*

Eyes bare, Wing with pale and dark spot.................................................................. 2

2 (1) $m_1$, $m_2$, $m_4$ cell with pale........................................................................................ 3

**Table 2. Measurements of distinguishing characteristics of *Culicoides* (*Sinocoides*) in China (Female).**

| species | wing length (mm) | CR | AR | PR | mandible teeth | antennal sensilla pattern |
|---|---|---|---|---|---|---|
| *C. anthropophygas* | 1.35 | 0.61 | 1.29 | 2.50 | 14 | 3, 9~15 |
| *C. hamiensis* | 1.18 | 0.57 | 0.96 | 2.50 | 18 | 3~14 |
| *C. jiangchengensis* | 1.08 | 0.50 | 1.13 | 2.63 | 17 | 3, 11~15 |
| *C. jinghongensis* | 1.08 | 0.54 | 0.96 | 1.70 | 12 | 3, 7~10 |
| *C. kongmiaoensis* | 1.18 | 0.53 | 1.59 | 2.42 | 13 | 3~15 |
| *C. multifarious* | 0.97 | 0.55 | 0.98 | 2.44 | 13 | 3, 8~10 |
| *C. malipoensis* | 1.35 | 0.70 | 1.05 | 2.90 | 9 | 3, 11~15 |
| *C. nanniwanensis* | 1.05 | 0.57 | 1.18 | 2.33 | 15 | 3, 11~15 |
| *C. opertus* | 1.13 | 0.59 | 1.20 | 2.50 | 18 | 3, 11~14 |
| *C. pungobovis* | 0.78 | 0.58 | 0.82 | 2.00 | 17 | 3, 8~10 |

Note CR: Costal ratio; AR: Antennal ratio; PR: Proboscis ratio.

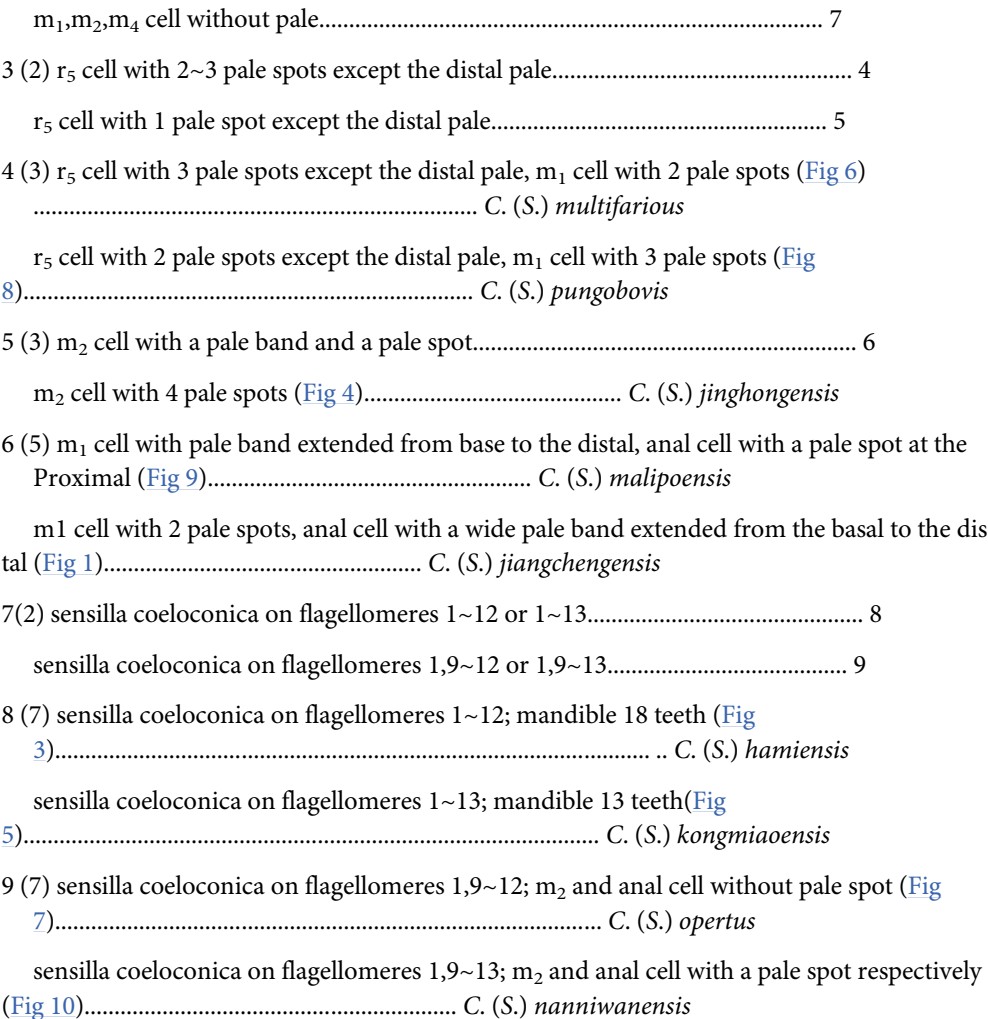

 $m_1$,$m_2$,$m_4$ cell without pale....................................................................................... 7

3 (2) $r_5$ cell with 2~3 pale spots except the distal pale...................................................... 4

 $r_5$ cell with 1 pale spot except the distal pale......................................................... 5

4 (3) $r_5$ cell with 3 pale spots except the distal pale, $m_1$ cell with 2 pale spots (Fig 6)
 ........................................................................... *C.* (*S.*) *multifarious*

 $r_5$ cell with 2 pale spots except the distal pale, $m_1$ cell with 3 pale spots (Fig
8)........................................................................... *C.* (*S.*) *pungobovis*

5 (3) $m_2$ cell with a pale band and a pale spot................................................................ 6

 $m_2$ cell with 4 pale spots (Fig 4)............................................ *C.* (*S.*) *jinghongensis*

6 (5) $m_1$ cell with pale band extended from base to the distal, anal cell with a pale spot at the
 Proximal (Fig 9)..................................................... *C.* (*S.*) *malipoensis*

 m1 cell with 2 pale spots, anal cell with a wide pale band extended from the basal to the distal (Fig 1)..................................................... *C.* (*S.*) *jiangchengensis*

7(2) sensilla coeloconica on flagellomeres 1~12 or 1~13............................................. 8

 sensilla coeloconica on flagellomeres 1,9~12 or 1,9~13....................................... 9

8 (7) sensilla coeloconica on flagellomeres 1~12; mandible 18 teeth (Fig
 3)............................................................................................ .. *C.* (*S.*) *hamiensis*

 sensilla coeloconica on flagellomeres 1~13; mandible 13 teeth(Fig
5)............................................................................................ *C.* (*S.*) *kongmiaoensis*

9 (7) sensilla coeloconica on flagellomeres 1,9~12; $m_2$ and anal cell without pale spot (Fig
 7)....................................................................................... *C.* (*S.*) *opertus*

 sensilla coeloconica on flagellomeres 1,9~13; $m_2$ and anal cell with a pale spot respectively
(Fig 10).............................................................. *C.* (*S.*) *nanniwanensis*

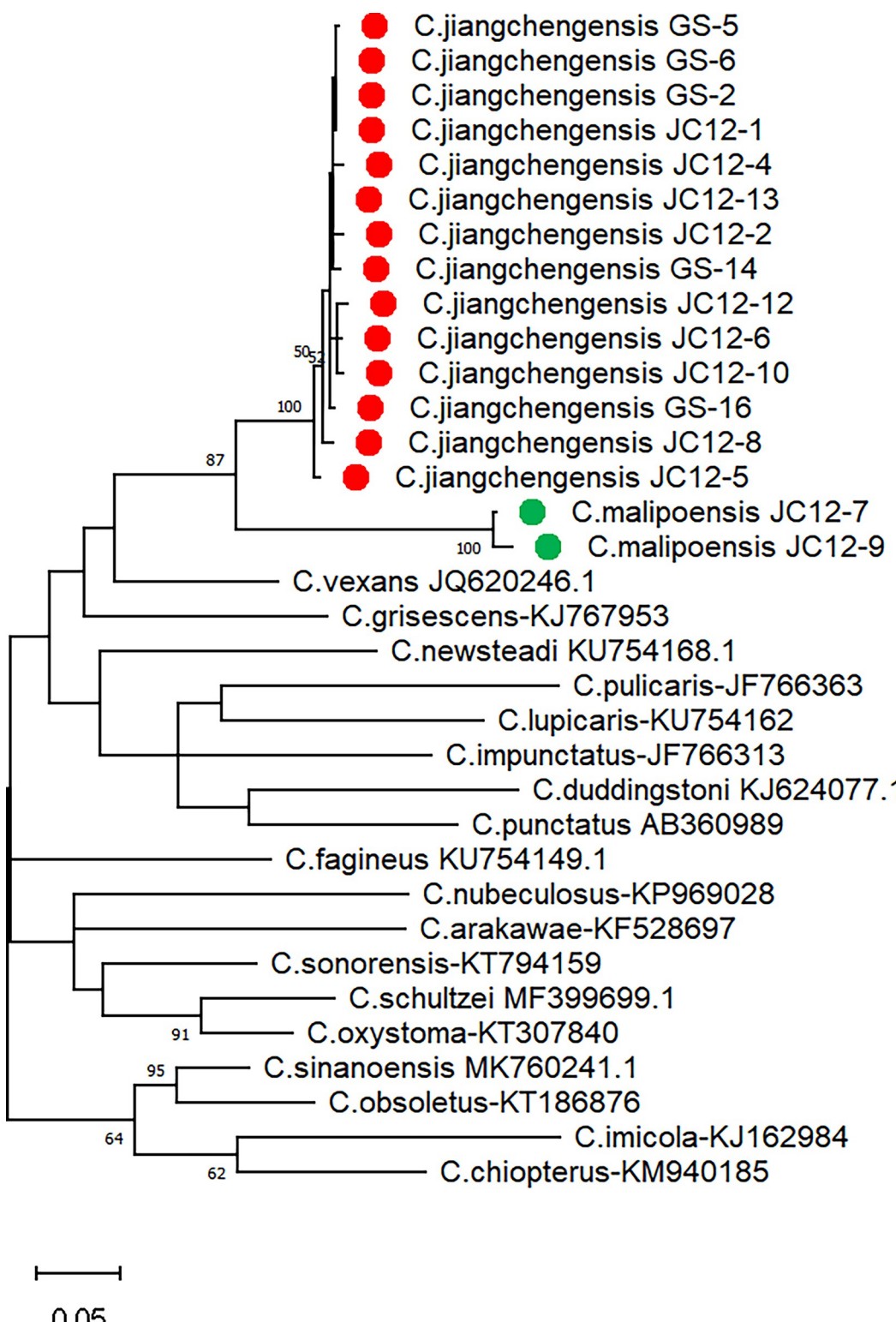

**Fig 11. ML phylogenetic trees of COI gene nucleotide sequences of *C.* (*Sinocoides*) *jiangchengensis* from Jiangcheng County and Gongshan County, Yunnan Province, China using MAGE-X.** The best DNA substitution model is GTR+G +I, and the trees were drawn using njplot 2.4 with the bootstrap value based on 1,000 replications.

## Supporting information

**S1 File.**
(DOCX)

## Author Contributions

**Conceptualization:** Guoping Liu, Jinglin Wang.

**Data curation:** Jinglin Wang.

**Formal analysis:** Yuwen He, Jinglin Wang.

**Funding acquisition:** Jinglin Wang.

**Investigation:** Yuwen He, Jinxin Meng, Zhao Li, Taoying Yu, Laxi Zhang, Dongmei Wang, Jinglin Wang.

**Methodology:** Yuwen He, Jinxin Meng, Nan Li.

**Project administration:** Jinglin Wang.

**Resources:** Yuwen He, Jinglin Wang.

**Software:** Jinglin Wang.

**Supervision:** Jinglin Wang.

**Writing – original draft:** Guoping Liu, Jinglin Wang.

**Writing – review & editing:** Jinglin Wang.

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
