## [Decision Letter · Decision Letter 0]

22 Dec 2022

PONE-D-22-32078Culicoides jiangchengensis, a new species of the subgenus Sinocoides (Diptera, Ceratopogonidae) based on integrative taxonomy from ChinaPLOS ONE

Dear Dr. Wang,

Thank you for submitting your manuscript to PLOS ONE. After careful consideration, we feel that it has merit but does not fully meet PLOS ONE’s publication criteria as it currently stands. Therefore, we invite you to submit a revised version of the manuscript that addresses the points raised during the review process.

We look forward to receiving your revised manuscript.

Kind regards,

Bi-Song Yue, Ph.D

Academic Editor

PLOS ONE

“This work is supported by grants from the Basic Research Projects of Yunnan Province (grant no. 2019FA015 and 202201AS070062), National Natural Science Foundation of China (grant no. 32260896 and 81960605), Scientific and technological innovation team construction project of Kunming Medical University (grant no. CXTD202111), Projects funded by the central government to guide local scientific and Technological Development (202207AB110006), and Yunnan Chenggong expert workstation (202005AF150034). The funders had no role in the study design, data collection and analysis, decision to publish, or preparation of the manuscript.”

“This work is supported by grants from the Basic Research Projects of Yunnan Province (grant no. 2019FA015 and 202201AS070062), National Natural Science Foundation of China (grant no. 32260896 and 81960605), Scientific and technological innovation team construction project of Kunming Medical University (grant no. CXTD202111), Projects funded by the central government to guide local scientific and Technological Development (202207AB110006), and Yunnan Chenggong expert workstation (202005AF150034). The funders had no role in the study design, data collection and analysis, decision to publish, or preparation of the manuscript.”

5. Please take this opportunity to be sure you have met all of our guidelines for new species. For proper registration of a new zoological taxon, we require two specific statements to be included in your manuscript.

a. In the Results section, the globally unique identifier (GUID), currently in the form of a Life Science Identifier (LSID), should be listed under the new species name, for example:

Anochetus boltoni Fisher sp. nov. urn:lsid:zoobank.org:act:B6C072CF-1CA6-40C7-8396-534E91EF7FBB

Another LSID for the manuscript itself should also appear within the Nomenclature statement. You will need to contact Zoobank (zoobank.org/About) to obtain a GUID (LSID). You should receive one LSID for your manuscript and a separate, unique LSID for the new species.

b. Please also insert the following text into the Methods section, in a sub-section to be called "Nomenclatural Acts":

The electronic edition of this article conforms to the requirements of the amended International Code of Zoological Nomenclature, and hence the new names contained herein are available under that Code from the electronic edition of this article. This published work and the nomenclatural acts it contains have been registered in ZooBank, the online registration system for the ICZN. The ZooBank LSIDs (Life Science Identifiers) can be resolved and the associated information viewed through any standard web browser by appending the LSID to the prefix "" ext-link-type="uri" xlink:type="simple">http://zoobank.org/". The LSID for this publication is: urn:lsid:zoobank.org:pub: XXXXXXX. The electronic edition of this work was published in a journal with an ISSN, and has been archived and is available from the following digital repositories: PubMed Central, LOCKSS [author to insert any additional repositories].

All PLOS ONE articles are deposited in PubMed Central and LOCKSS. If your institute, or those of your co-authors, has its own repository, we recommend that you also deposit the published online article there and include the name in your article.

Following a recent ruling by the International Commission on Zoological Nomenclature, electronic journals are now a valid format for publication of new zoological taxa. In order to ensure the valid publication of your new species, please be sure to include the updated version of Nomenclatural Acts (above). A complete explanation of our guidelines for publishing new species can be found on our website: http://www.plosone.org/static/guidelines#zoological.

Reviewers' comments:

Reviewer's Responses to Questions

**Comments to the Author**

1. Is the manuscript technically sound, and do the data support the conclusions?

Reviewer #1: Yes

Reviewer #2: Yes

Reviewer #3: Partly

Reviewer #4: Yes

2. Has the statistical analysis been performed appropriately and rigorously? 

Reviewer #1: Yes

Reviewer #2: N/A

Reviewer #3: N/A

Reviewer #4: No

3. Have the authors made all data underlying the findings in their manuscript fully available?

Reviewer #1: Yes

Reviewer #2: Yes

Reviewer #3: Yes

Reviewer #4: Yes

4. Is the manuscript presented in an intelligible fashion and written in standard English?

Reviewer #1: Yes

Reviewer #2: No

Reviewer #3: No

Reviewer #4: Yes

5. Review Comments to the Author

Reviewer #1: Comments:

This manuscript described a new biting midge species from China along with checklist and keys to adult female of the subgenus Sinocoides. The mitochondrial COI sequences were also provided for the new species. Therefore, information present in this study is useful and interesting. Following are my comments and suggestions:

- Abstract, I suggested to modify the first sentence as “Biting midges of the genus Culicoides are medical and veterinary important because ....”

- Abstract, 3rd line, change “The” to “the”

- Abstract, 4th line, remove “both sexes”

- Abstract, 5th line, remove “in detail”

- Abstract, second page, 2nd and 3rd lines, Diptera, Ceratopogonidae should not italic, and remove underline from Culicoides.

- Introduction, 1st line, add “causing agents” after disease, vectors do not carry disease, they carrying disease causing agents.

- Materials and methods, more details of PCR should be provided e.g. primers, reaction conditions. Also, details of PCR product purification and sequencing must be provided. In addition, the details of MEGA-X software analysis must be provided e.g. analysis what? criteria setting for data analysis?

- Diagnosis, I am wondering why “only for Culicoides species in Yunnan”? Why not for Culicoides in other regions? Please explain this.

- Discussion, it will be very useful to add more details of morphological characteristics that a new species is different from previously described species. For examples, “C. jiangchegensis 1 ) without sensory pit in 3rdsegment of palpus; 2 ) sensilla coeloconica on flagellomere 1, 9-13; 3 ) m1 and m2 cell of the wings with pale spots, which were different from the other eight species of culicoides in subgenus Sinocoides, such as C. anthropophygas (with sensory pit in 3rd segment of palpus???, sensilla coeloconica on flagellomere ?????, m1 and m2 cell of the wings with ???? spots,)...” Details of morphological comparison like this should also provide for other species, particularly for morphologically closely related species, C. malipoensis.

- Figure 11, phylogenetic tree, the outgroup species should be included in the phylogenetic analysis. Bootstrap values lower than 50% should not show on the figure.

- English should be check again by native speaker.

Reviewer #2: I think that this type of studies should be published more in taxonomy journals. In this study, the existence of a new Culicoides species, its place in classification and its morphological features were described. Therefore, I think this research paper is not suitable for PLOS ONE.

Reviewer #3: The submitted manuscript provides very interesting and significant results. However, the text is not checked very carefully and it consists of many small technical mistakes. English should be revised. Structure of the manuscript should be modified in the direction to be simple to follow and to read. Descriptions should be given within the regular English structure of sentences. Verbs are missing in the description text.

Specific comments are given below:

Abstract

“blood-sucking feeding habits”: it is not the habit but the blood-feeding regime or haematophagous feeding regime.

The morphological: small letter

culicoides: capital letters. Culicoides should be in italic. If authors wanted to write in English than is biting midges

Genetically, C. jiangchengensis most closely related to C. malipoensis: missing verb ... are most closely related to....

Diptera: Ceratopogonidae: this should not be in italic.

Introduction

that transmit arboviruses .... is enough

This cannot be cited this way “Liu et al. (1990, 1996, 2003)” because first is Liu and Yu and other two are Liu et al.

Same comment as previous for Liu et al. (2006, 2011) and Wu et al. (2018).

Material and methods

august big lletter

Diagnosis: this text should be significantly improved because in this way is very difficult to follow and to read.

Page 13: Please delete redundant space

C.jiangchengensis: here is missing space

Table 2. Abbreviations should be given below the table or in the title

Figure titles are missing spaces

Reviewer #4: This manuscript reports a new species of the biting midge subgenus Sinocoides of the genus Culicoides by using integrative approaches from morphological and molecular evidence. The authors provide detail description and illustrations of the morphological characters of the new species and other described species of the same subgenus. The identification appears to be accurate. In general, the manuscript is technically solid. I made some minor notes in the pdf file of the manuscript. Besides, I suggest the authors provide more details about the molecular phylogenetic analysis. For example, in Material and Methods, please provide how the sampling was designed and what the analytical methods they used for the phylogenetic analysis.

6. PLOS authors have the option to publish the peer review history of their article (what does this mean?). If published, this will include your full peer review and any attached files.

Reviewer #1: No

Reviewer #2: No

Reviewer #3: No

Reviewer #4: No

---

## [Author Response · Author response to Decision Letter 0]

21 Feb 2023

Dear Bi-Song Yue, Ph.D

Thank you for your letter (on 23th December 2022) and for the referees’ thoughtful comments concerning our manuscript entitled " Culicoides jiangchengensis, a new species of the subgenus Sinocoides (Diptera, Ceratopogonidae) based on integrative taxonomy from China” (PONE-D-22-32078).

We have studied reviewer comments carefully and have also addressed the comments as described below and made corrections on the revised version that we hope meet with their approval. 

Thank you very much for your consideration.

Sincerely yours,

Jinglin Wang, Ph.D.

Yunnan Tropical and Subtropical Animal Viral Disease Laboratory, 

Yunnan Animal Science and Veterinary Institute, Kunming, Yunnan province. 

Qinglongshan, Jindian, Pang Long District, Kunming 650224, P.R. China 

Tel: 86-871-6501-5606 

Fax: 86-871-6501-0043 

E-mail: wangjl107@163.com

Response

1.When submitting your revision, we need you to address these additional requirements.

Answer: I have carefully modified the format of the article according to the PLOS ONE's style requirements, carefully checked grant numbers, deleted the Funding Statement in the Acknowledgments Section, moved the ethics statement to the Methods section, and properly registered a new zoological taxon of the Culicoides. 

Funding Statement: This work is supported by grants from the Basic Research Projects of Yunnan Province (grant no. 2019FA015 and 202201AS070062), National Natural Science Foundation of China (grant no. 32260896 and 81960605), Scientific and technological innovation team construction project of Kunming Medical University (grant no. CXTD202111), Projects funded by the central government to guide local scientific and Technological Development (202207AB110006), and Yunnan Chenggong expert workstation (202005AF150034). The funders had no role in the study design, data collection and analysis, decision to publish, or preparation of the manuscript.

Reviewer #1:

1.- Abstract, I suggested to modify the first sentence as “Biting midges of the genus Culicoides are medical and veterinary important because ....”

Answer: According to the comment, we have modify the first sentence in the new manuscript。

2.- Abstract, 3rd line, change “The” to “the”

- Abstract, 4th line, remove “both sexes”

- Abstract, 5th line, remove “in detail”

- Abstract, second page, 2nd and 3rd lines, Diptera, Ceratopogonidae should not italic, and remove underline from Culicoides.

Answer: According to the comment, these errors in the abstract have been modified in the new manuscript.

3.- Introduction, 1st line, add “causing agents” after disease, vectors do not carry disease, they carrying disease causing agents.

Answer: According to the comment, we have revised this sentence in the new manuscript.

3.- Materials and methods, more details of PCR should be provided e.g. primers, reaction conditions. Also, details of PCR product purification and sequencing must be provided. In addition, the details of MEGA-X software analysis must be provided e.g. analysis what? criteria setting for data analysis?

Answer: According to the comment, we have added the above contents in the new manuscript.

5.- Diagnosis, I am wondering why “only for Culicoides species in Yunnan”? Why not for Culicoides in other regions? Please explain this.

Answer: we have deleted the word “in Yunnan” in the new manuscript.

6.- Discussion, it will be very useful to add more details of morphological characteristics that a new species is different from previously described species. For examples, “C. jiangchegensis 

1 ) without sensory pit in 3rdsegment of palpus; 

2 ) sensilla coeloconica on flagellomere 1, 9-13; 

3 ) m1 and m2 cell of the wings with pale spots, which were different from the other eight species of culicoides in subgenus Sinocoides, such as C. anthropophygas (with sensory pit in 3rd segment of palpus???, sensilla coeloconica on flagellomere ?????, m1 and m2 cell of the wings with ???? spots,)...” Details of morphological comparison like this should also provide for other species, particularly for morphologically closely related species, C. malipoensis.

Answer: More details of morphological characteristics that a new species is different from previously described species have been described in the section Remarks. 

7.- Figure 11, phylogenetic tree, the outgroup species should be included in the phylogenetic analysis. Bootstrap values lower than 50% should not show on the figure.

Answer: The phylogenetic tree was constructed based on the COI gene sequences of C. jiangchengensis, C. malipoensis from Jiangcheng County and Gongshan County, Yunnan Province and another 24 species of Culicoides. These other 24 species of Culicoides can be used as the outgroups species. According to the comment, bootstrap values lower than 50% should not show on the figure 11.

8.- English should be check again by native speaker.

Answer: We have modified the English Language in the new manuscript. The English in this document has been checked by at least two professional editors, both native speakers of English. For a certificate, please see: 

http://www.textcheck.com/certificate/WoMsXU

9.Reviewer #2: I think that this type of studies should be published more in taxonomy journals. In this study, the existence of a new Culicoides species, its place in classification and its morphological features were described. Therefore, I think this research paper is not suitable for PLOS ONE.

Answer: I agree with the Reviewer #2 that our study is about taxonomy. However, considering that PLOS ONE is a high-level comprehensive magazine, we think that our study suitable for PLOS ONE, so we submitted the manuscript into the journal.

10.Reviewer #3: The submitted manuscript provides very interesting and significant results. However, the text is not checked very carefully and it consists of many small technical mistakes. English should be revised. Structure of the manuscript should be modified in the direction to be simple to follow and to read. Descriptions should be given within the regular English structure of sentences. Verbs are missing in the description text.

Answer: We have read a lot of literature on the morphological characteristics of animals. When describing the morphological characteristics of animal species, verbs are missing in the description text so that readers can quickly capture the information of animal morphological characteristics. The English in this manuscript has been checked by at least two professional editors, both native speakers of English. For a certificate, please see: 

http://www.textcheck.com/certificate/WoMsXU

Specific comments are given below:

11.Abstract

“blood-sucking feeding habits”: it is not the habit but the blood-feeding regime or haematophagous feeding regime.

Answer: According to the comment, we have revised it in the new manuscript.

12.The morphological: small letter

culicoides: capital letters. Culicoides should be in italic. If authors wanted to write in English than is biting midges

Genetically, C. jiangchengensis most closely related to C. malipoensis: missing verb ... are most closely related to....

Diptera: Ceratopogonidae: this should not be in italic.

Answer: According to the comment, we have revised them in the new manuscript.

13.Introduction

that transmit arboviruses .... is enough

This cannot be cited this way “Liu et al. (1990, 1996, 2003)” because first is Liu and Yu and other two are Liu et al.

Same comment as previous for Liu et al. (2006, 2011) and Wu et al. (2018).

Answer: According to the comment, we have revised them in the new manuscript.

14.Material and methods

august big lletter

Diagnosis: this text should be significantly improved because in this way is very difficult to follow and to read.

Answer: According to the comment, we have revised them in the new manuscript.

15.Page 13: Please delete redundant space

C.jiangchengensis: here is missing space

Table 2. Abbreviations should be given below the table or in the title

Figure titles are missing spaces

Answer: According to the comment, we have revised them in the new manuscript.

16.Reviewer #4: This manuscript reports a new species of the biting midge subgenus Sinocoides of the genus Culicoides by using integrative approaches from morphological and molecular evidence. The authors provide detail description and illustrations of the morphological characters of the new species and other described species of the same subgenus. The identification appears to be accurate. In general, the manuscript is technically solid. I made some minor notes in the pdf file of the manuscript. Besides, I suggest the authors provide more details about the molecular phylogenetic analysis. For example, in Material and Methods, please provide how the sampling was designed and what the analytical methods they used for the phylogenetic analysis.

Answer: According to the comment, we provide more details about the molecular phylogenetic analysis in Material and Methods. some minor notes in the pdf file of the manuscript have revised in the new manuscript.

---

## [Decision Letter · Decision Letter 1]

17 Apr 2023

PONE-D-22-32078R1Culicoides jiangchengensis, a new species of the subgenus Sinocoides (Diptera, Ceratopogonidae) based on integrative taxonomy from ChinaPLOS ONE

Dear Dr. Wang,

Thank you for submitting your manuscript to PLOS ONE. After careful consideration, we feel that it has merit but does not fully meet PLOS ONE’s publication criteria as it currently stands. Therefore, we invite you to submit a revised version of the manuscript that addresses the points raised during the review process.

In your revision, please address the minor stylistic corrections indicated by reviews 3.

We look forward to receiving your revised manuscript.

Kind regards,

Ulrike Gertrud Munderloh, Ph.D.

Academic Editor

PLOS ONE

Journal Requirements:

Reviewers' comments:

Reviewer's Responses to Questions

**Comments to the Author**

1. If the authors have adequately addressed your comments raised in a previous round of review and you feel that this manuscript is now acceptable for publication, you may indicate that here to bypass the “Comments to the Author” section, enter your conflict of interest statement in the “Confidential to Editor” section, and submit your "Accept" recommendation.

Reviewer #1: All comments have been addressed

Reviewer #2: All comments have been addressed

Reviewer #3: (No Response)

2. Is the manuscript technically sound, and do the data support the conclusions?

Reviewer #1: Yes

Reviewer #2: Partly

Reviewer #3: Yes

3. Has the statistical analysis been performed appropriately and rigorously? 

Reviewer #1: Yes

Reviewer #2: N/A

Reviewer #3: N/A

4. Have the authors made all data underlying the findings in their manuscript fully available?

Reviewer #1: Yes

Reviewer #2: Yes

Reviewer #3: Yes

5. Is the manuscript presented in an intelligible fashion and written in standard English?

Reviewer #1: Yes

Reviewer #2: Yes

Reviewer #3: Yes

6. Review Comments to the Author

Reviewer #1: (No Response)

Reviewer #2: This is a taxonomical study and not suitable for plos one journal. Thetefore it can be published in another associated journal.

Reviewer #3: After I checked again the manuscript I could conclude that authors accepted almost all suggestions and improved the manuscript. Their findings can be considered as significant contribution to the science.

Here are some minor corrections that were missed to be done:

Again Diptera and Ceratopogonidae should not be in italic.

Ceratopogonidae transmit viruses and not one virus. Please correct that in the first sentence of introduction.

Please correct “china” to capital letter

Dallas et al. (2003) should all go into brackets.

culicoides should be in capital letter

7. PLOS authors have the option to publish the peer review history of their article (what does this mean?). If published, this will include your full peer review and any attached files.

Reviewer #1: No

Reviewer #2: No

Reviewer #3: No

---

## [Author Response · Author response to Decision Letter 1]

8 May 2023

Dear Ulrike Gertrud Munderloh, Ph.D

Thank you for your letter (on 17th April 2023) and for the referees’ thoughtful comments concerning our manuscript entitled " Culicoides jiangchengensis, a new species of the subgenus Sinocoides (Diptera, Ceratopogonidae) based on integrative taxonomy from China” (PONE-D-22-32078R1).

We have studied reviewer comments carefully and have also addressed the comments as described below and made corrections on the revised version that we hope meet with their approval. 

Thank you very much for your consideration.

Sincerely yours,

Jinglin Wang, Ph.D.

Yunnan Tropical and Subtropical Animal Viral Disease Laboratory, 

Yunnan Animal Science and Veterinary Institute, Kunming, Yunnan province. 

Qinglongshan, Jindian, Pang Long District, Kunming 650224, P.R. China 

Tel: 86-871-6501-5606 

Fax: 86-871-6501-0043 

E-mail: wangjl107@163.com

Response

1.Again Diptera and Ceratopogonidae should not be in italic.

Answer: According to the comment, we have modify them in the new manuscript

2.Ceratopogonidae transmit viruses and not one virus. Please correct that in the first sentence of introduction.

Answer: According to the comment, we have modify the first sentence of introduction in the new manuscript.

3.Please correct “china” to capital letter

Answer: According to the comment, we have corrected “china” to capital letter in the new manuscript.

4.Dallas et al. (2003) should all go into brackets.

Answer:According to the comment, we have put “Dallas et al. ( 2003 )” all in brackets.

5.culicoides should be in capital letter

Answer: According to the comment, we have revised it in the new manuscript.

---

## [Editor Report · Decision Letter 2]

2 Jun 2023

Culicoides jiangchengensis, a new species of the subgenus Sinocoides (Diptera, Ceratopogonidae) based on integrative taxonomy from China

PONE-D-22-32078R2

Dear Dr. Wang,

We’re pleased to inform you that your manuscript has been judged scientifically suitable for publication and will be formally accepted for publication once it meets all outstanding technical requirements.

Kind regards,

Ulrike Gertrud Munderloh, Ph.D.

Academic Editor

PLOS ONE
---

## [Editor Report · Acceptance letter]

4 Jul 2023

PONE-D-22-32078R2 

*Culicoides jiangchengensis*, a new species of the subgenus *Sinocoides* (Diptera, Ceratopogonidae) based on integrative taxonomy from China 

Dear Dr. Wang:

I'm pleased to inform you that your manuscript has been deemed suitable for publication in PLOS ONE. Congratulations! Your manuscript is now with our production department. 

Kind regards, 

on behalf of

Dr. Ulrike Gertrud Munderloh 

Academic Editor

PLOS ONE